# Platelets Modulate Leukocyte Population Composition Within Perivascular Adipose Tissue

**DOI:** 10.3390/ijms26041625

**Published:** 2025-02-14

**Authors:** Adam Corken, Tiffany Weinkopff, Elizabeth C. Wahl, James D. Sikes, Keshari M. Thakali

**Affiliations:** 1Department of Pediatrics, University of Arkansas for Medical Sciences, Little Rock, AR 72202, USA; alcorken@uams.edu; 2Arkansas Children’s Research Institute, Little Rock, AR 72202, USA; wahlec@archildrens.org (E.C.W.); sikesjd@archildrens.org (J.D.S.); 3Department of Microbiology and Immunology, University of Arkansas for Medical Sciences, Little Rock, AR 72205, USA; tsweinkopff@uams.edu

**Keywords:** perivascular adipose tissue, platelets, leukocytes, monocytes, macrophages, Western diet, obesity

## Abstract

Perivascular adipose tissue (PVAT) regulates vascular tone and is composed of adipocytes and several leukocyte subpopulations. Diet can modify PVAT function, as obesogenic diets cause morphological changes to adipocytes and skew the leukocyte phenotype, leading to PVAT dysregulation and impaired vasoregulation. Of note, platelets, the clot-forming cells, also modulate many facets of leukocyte activity, such as tissue infiltration and polarity. We aimed to determine whether platelets regulate the leukocyte populations residing within PVAT. Male C57Bl/6J mice were fed a Western diet (30% kcal sucrose, 40% kcal fat, 8.0% sodium) to develop obesogenic conditions for PVAT leukocyte remodeling. Diet was either administered acutely (2 weeks) or extended (8 weeks) to gauge the length of challenge necessary for remodeling. Additionally, platelet depletion allowed for the assessment of platelet relevance in PVAT leukocyte remodeling. Abdominal PVAT (aPVAT) and thoracic PVAT (tPVAT) were then isolated and leukocyte composition evaluated by flow cytometry. Compared to control, Western diet alone did not significantly impact PVAT leukocyte composition for either diet length. Platelet depletion, independent of diet, significantly disrupted PVAT leukocyte content with monocytes/macrophages most impacted. Furthermore, tPVAT appeared more sensitive to platelet depletion than aPVAT, providing novel evidence of platelet regulation of leukocyte composition within PVAT depots.

## 1. Introduction

PVAT is the adipocyte-rich tissue surrounding the blood vessels of the vascular network and serves as the fourth vessel layer due to its proximal location to the tunica adventitia [1,2]. PVAT functions to maintain vascular tone via the secretion of soluble mediators that regulate the contractile state of the vascular smooth muscle (VSM) layer [3]. Under physiological conditions, PVAT promotes an anti-contractile state via paracrine mediators, thereby maintaining total peripheral resistance and blood pressure at normal healthy levels [4]. However, pathophysiological conditions, such as obesity and hypertension, prompt changes in PVAT function, resulting in the reduction of the anti-contractile properties of PVAT, eliciting VSM constriction and blood pressure elevation [5,6,7,8,9,10]. This shift in PVAT is due to the remodeling of the adipocyte and leukocyte populations within the tissue, which itself is caused by an increased consumption of calorie-dense, nutrient-poor, processed foods [11,12]. Normally, the adipocytes and leukocytes work in concert to maintain a relaxed, anti-inflammatory vascular environment, but improper nutrition promotes an obesogenic state, resulting in PVAT adipocytes aiding in the uptake of increased circulating lipids. The enhanced lipid uptake increases the cells’ surface-area-to-volume ratio, thereby generating a hypoxic, pro-inflammatory adipocyte phenotype [13]. Likewise, this diet-induced obesogenic state compels a switch in the resident leukocyte activation states from a suppressive environment dictated by regulatory T cells in favor of a pro-inflammatory environment with Th1, Th17 and CD8+ cells [14,15,16]. Similarly, a change in macrophage polarization from an M2 to an M1 activation state was postulated as indicative of a pro-inflammatory environment facilitating tissue disruption, although other studies implicated M2 macrophages in PVAT dysregulation as well [17,18,19,20,21]. As such, a defining hallmark of obesogenic dietary remodeling of PVAT is the presence of pro-inflammatory leukocytes.

Platelets are blood cells that principally mediate clot formation to maintain vascular integrity following vessel injury in a process known as hemostasis [22]. Recently, the platelet’s purview has extended to include the initiation and maintenance of the inflammatory response [23,24]. Platelets demonstrate a versatile range of immune functions through the direct and indirect regulation of leukocyte activity. For instance, platelets associate with a number of immune cells in circulation, most notably with monocytes and neutrophils [25]. This interplay between platelets and leukocytes modulates the latter’s functional and secretory activity [26,27,28]. Moreover, platelets govern the phenotype of monocytes and their derivative macrophages by influencing the macrophage polarization or activation status [29,30]. Platelets also facilitate the extravasation of immune cells into neighboring tissues by directly anchoring leukocytes to the vessel wall or via the secretion of chemokines [31,32,33,34]. Because platelets can influence the recruitment and polarization of leukocytes under various inflammatory conditions, we hypothesized that platelets may play a role in the remodeling of leukocytes within PVAT in obesity. Thus, platelets represent an unexplored avenue by which an obesogenic state directs the reorganization of leukocytes residing within PVAT. In this study, by depleting circulating platelets, we examined the role platelets play in altering leukocyte populations within PVAT depots in mice fed a control or “Western” diet (high in sucrose, fat and salt). The results highlight the importance of platelets as key orchestrators for several leukocyte subsets within PVAT depots.

## 2. Results

### 2.1. Leukocyte PVAT Composition with Acute Western Diet Feeding

To establish obesogenic conditions to investigate PVAT remodeling, mice were fed a Western diet and compared with a control diet. The changes in total body mass and lean/fat mass composition are outlined in Appendix A. The initial classification of leukocytes within aPVAT and tPVAT depots began with the identification of myeloid cells, as well as two lymphocyte subsets including B and T cells. Within aPVAT, the B cell population was the most abundant of PVAT leukocytes, with T cells displaying the next-highest proportion and myeloid cells comprising the lowest abundance (Figure 1A). Similar B cell content was observed in tPVAT, while T cells and myeloid cells displayed more equal distributions (Figure 1B). Neither diet nor platelet depletion affected the contents of any of these broad leukocyte populations after 2 weeks of feeding.

### 2.2. T and Myeloid Cell Subsets Within PVAT Following Acute Feeding

The CD3+ T cell population can be further subdivided into CD4 and CD8 positive T cell subsets. As such, we visualized the distribution of CD4+ and CD8+ T cells within aPVAT (Figure 2A) and tPVAT (Figure 2B). In both PVAT depots, CD4+ T cells were generally found in a higher abundance than CD8+ cells. Interestingly, within tPVAT, both the CD4+ and CD8+ T cells comprised slightly smaller proportions of the total T cell population relative to what was observed in aPVAT. The identification of CD11b+ myeloid subsets revealed several differences between the groups in both aPVAT (Figure 2C) and tPVAT (Figure 2D). Amongst both PVAT depots, monocytes were the most abundant myeloid cell type by frequency, independent of the diet or platelet treatment after 2 weeks. In both PVAT depots, intact platelets paired with a Western diet significantly reduced the polymorphonuclear neutrophil (neutrophil) population, with platelet depletion trending toward increased neutrophil content independent of diet. In addition, we found that platelet depletion significantly increased eosinophil content in both depots in the mice administered a Western diet. Furthermore, within both aPVAT and tPVAT, platelet depletion alone trended toward reducing the macrophage population, where the mice that received a Western diet demonstrated significant decreases.

### 2.3. Platelet Modulation of Monocytes and Macrophages in PVAT After Acute Western Diet

A flow cytometric analysis of PVAT monocytes revealed that an intact platelet population in conjunction with control diet feeding significantly diminished Ly6C^lo^CX3CR1^hi^ classical and conversely elevated Ly6C^hi^CX3CR1^lo^ nonclassical monocytes in the aPVAT (Figure 3A). Ly6C^lo^CX3CR1^hi^ classical monocytes have been broadly defined as inflammatory monocytes and Ly6C^hi^CX3CR1^lo^ nonclassical monocytes have been broadly defined as patrolling monocytes [35,36]. Likewise, platelet depletion with Western diet feeding reduced the nonclassical monocyte population in aPVAT while not affecting classical monocytes compared with both control diet groups. Within tPVAT, platelet depletion—independent of diet—increased the proportion of classical and simultaneously reduced nonclassical monocytes (Figure 3B). Additionally, the control diet cohort with an intact platelet population had significantly elevated nonclassical monocyte levels within tPVAT relative to the other groups. Thus, within tPVAT specifically, the removal of the platelet population was a more significant driver than diet in terms of influencing monocytes. Upon analysis of the macrophages, neither diet nor platelet treatment influenced macrophage polarization within aPVAT based on using iNOS as a surrogate for M1 activation and CD206 as a surrogate for M2 activation (Figure 3C) [37,38,39,40]. Despite this, platelet depletion significantly increased the abundance of M1 macrophages in tPVAT, albeit there was a low percentage of M1 macrophages in general (Figure 3D). Additionally, we found that M2 macrophages dominated both PVAT depots, independent of the diet or platelet treatment.

### 2.4. PVAT Leukocyte Content After Extended Western Diet Feeding

Extending the feeding duration to 8 weeks resulted in body mass accumulation for all the groups, independent of the diet or platelet treatment (Appendix A). Additionally, all groups increased their fat mass and reduced their lean mass, independent of diet with extended feeding (Appendix A). An 8-week feeding duration paired with varying platelet treatments resulted in disturbances in the myeloid, B and T cell populations in PVAT that were not present with acute feeding. Within aPVAT, platelet depletion coupled with a Western diet significantly increased, albeit slightly, the frequency of T cells compared with all the other groups (Figure 4A). Additionally, intact platelets slightly but significantly reduced the B cell frequency when paired with a Western diet compared with the platelet-depleted groups. In tPVAT, platelet depletion independent of diet significantly reduced the myeloid cell population (Figure 4B), and when matched with a Western diet, increased the T cell population similar to that of aPVAT. Intact platelets in conjunction with a Western diet also reduced the B cell population similar to aPVAT.

### 2.5. Myeloid Cell Subsets Shifted with Extended Western Diet Feeding and Platelet Depletion

Previously, 2 weeks of feeding revealed that diet and platelet depletion significantly altered myeloid cell content while having negligible effects on T cells. By extending the feeding duration to 8 weeks, we observed perturbations in the T cell subset, as well as in myeloid cells. A Western diet paired with platelet depletion increased CD4+ and reduced CD8+ T cells in aPVAT (Figure 5A). Within tPVAT, Western diet and platelet depletion similarly increased CD4+ cell content while having no discernable effect on CD8+ cells (Figure 5B). We noted that with 8 weeks of feeding, aPVAT (Figure 5C) and tPVAT (Figure 5D) both displayed a trend toward a reduction in eosinophil content when the platelet population was maintained with a control diet, while platelet depletion paired with the control diet facilitated an increase in this population in aPVAT only. Additionally, platelet depletion combined with a control diet trended toward reducing the neutrophil content in aPVAT.

### 2.6. Platelets Influenced Monocyte and Macrophage Subsets with Extended Feeding

With the 2 weeks of feeding, platelet depletion independently impacted monocyte subset distributions in tPVAT while having less of an effect on aPVAT. However, increasing the duration of feeding to 8 weeks resulted in the emergence of platelet influence in aPVAT. When comparing the platelet-depleted groups to those with an intact platelet population, there was a weak trend toward fewer Ly6C^hi^CX3CR1^lo^ classical and more Ly6C^lo^CX3CR1^hi^ nonclassical monocytes in the absence of platelets (Figure 6A). Furthermore, platelet depletion with control diet significantly decreased classical and increased nonclassical monocytes relative to control groups in aPVAT. Similarly, in tPVAT, platelet depletion following 8 weeks of feeding reduced the percentage of the classical and increased the percentage of nonclassical monocytes, independently of diet (Figure 6B). Thus, the trend in platelet depletion was similar between PVAT depots, but with tPVAT displaying the most dramatic changes. Of note, in tPVAT, platelet depletion with extended feeding caused reduced classical and increased nonclassical monocyte frequencies, while with acute feeding, platelet depletion increased classical and decreased nonclassical monocytes. Similar to the monocyte subsets, platelets demonstrated an ability to modulate macrophage polarity with acute feeding, albeit in tPVAT depot alone. As such, we investigated the effect of extended feeding on macrophage activation status in the presence or absence of platelets. We found that platelet depletion, independent of diet, led to a significant reduction in M1 macrophage content for both aPVAT (Figure 6C) and tPVAT (Figure 6D) depots. Both depots similarly showed trends toward increased M2 macrophage content regardless of diet with platelet depletion. Overall, our analysis revealed most macrophages in aPVAT and tPVAT exhibited an M2 phenotype, as characterized by CD206 positivity, where very few macrophages (<1%) produced iNOS, a marker indicative of M1 activation. Interestingly, platelet depletion caused M1 macrophage reduction in tPVAT under extended feeding conditions, whereas under acute feeding conditions, platelet depletion increased the M1 macrophages.

## 3. Discussion

The purpose of our study was to determine the role of platelets in the process of PVAT-leukocyte remodeling in an obesogenic environment. To address this, we utilized both acute 2-week and extended 8-week feeding regimens. The aim of acute feeding was to determine whether PVAT leukocyte composition remodeled rapidly, while extended feeding mimicked remodeling under more chronic conditions. Moreover, we chose to investigate PVAT from the abdominal and thoracic segments of the aorta, as they are the most widely researched PVAT depots and have similar functional properties, though they differ subtly when comparing adipocyte morphology and color [41,42]. Furthermore, platelet depletion was conducted during the final 10 days of each feeding protocol to investigate whether platelets are involved in early PVAT remodeling (2 week feeding) or whether remodeling is a constant, ongoing process (8 weeks of feeding). And so, the observation of leukocyte disruption at either an early and/or later platelet-depleting intervention would additionally elucidate whether PVAT remodeling was a singularly affixed or an actively maintained process. Strictly dietary contributions to PVAT leukocyte composition were noted by comparing a vehicle-treated (platelet-maintained) control to Western diet groups.

Dietary influence emerged at 8 weeks of feeding as a Western diet elevated the percentage of T cells and reduced the percentage of neutrophils in aPVAT. Within tPVAT, Western diet increased classical and lowered nonclassical monocyte levels with extended feeding. These findings indicate that an obesogenic dietary stimulus disrupted PVAT leukocyte composition more as the diet duration increased. Surprisingly, we discovered that platelets, more so than diet, were a significant driver of PVAT leukocyte remodeling. We discovered that the removal of platelets generated rapid changes in PVAT leukocyte content, with the exact nature of the disruption dependent on the duration of dietary administration. While the effects of certain diets were augmented by platelet depletion, platelet-depleted groups more often displayed similar PVAT leukocyte profiles regardless of diet. These effects were clearly observed for multiple leukocyte subsets in aPVAT and tPVAT for both feeding durations. The loss of platelets resulted in an increased neutrophil content with 2 weeks of feeding and subsequently increased CD4+ T cell percentages after 8 weeks of feeding in tPVAT. The neutrophil trend was surprising, as previous literature indicates platelets largely enhance neutrophil recruitment, and so, we would have expected depletion of platelets to reduce neutrophil content [43]. Regardless, this indicates a temporal shift in the platelet’s calibration of PVAT leukocyte recruitment, as it favors limiting neutrophil infiltration with an acute feeding insult, then transitioning to restricting CD4+ T cell migration with a chronic feeding stimulus. Moreover, this targeted restriction of CD4+ T cell recruitment overlaps with prior studies that demonstrated platelet-mediated inhibition of CD4+ T cell recruitment [44].

Though platelets modified multiple leukocyte subsets within PVAT, their influence was most apparent when observing monocytes and macrophages. Prior studies highlighted the ability of platelets to enhance monocyte infiltration and increase resident macrophages within non-adipose tissues [45,46,47]. As such, platelets demonstrated a similar ability to direct monocytes/macrophages within PVAT, as platelet depletion diminished the total macrophage population in both PVAT depots and reduced monocytes in tPVAT alone with acute feeding. Platelets were also shown to dictate monocyte phenotype and macrophage polarization to either the M1 [30] or M2 [48] designation, depending on experimental circumstances. Accordingly, platelet depletion following acute feeding increased classical and decreased nonclassical monocyte populations, while also increasing the M1 macrophage phenotype in tPVAT. With extended feeding, platelet depletion reduced classical and increased nonclassical monocytes in tPVAT, in addition to diminishing M1 macrophage levels in both PVAT depots. This alludes to a temporal shift in platelet modulation of the classical monocyte population in relation to feeding duration within tPVAT, specifically as platelets diminish classical monocytes early but shift toward increasing this population as feeding is prolonged. Similarly, platelets hampered M1 macrophage recruitment with acute feeding, specifically in tPVAT, but reversed course and promoted M1 macrophage recruitment in both tPVAT and aPVAT with extended feeding. This robust platelet targeting of the monocyte/macrophage populations within PVAT relative to other leukocytes further highlights the intimate relationship between platelets and this leukocyte class previously described under various conditions [49,50,51,52,53,54,55].

Although this series of experiments represents a novel investigation into the previously undetermined role of platelet involvement in PVAT leukocyte composition, it was not without limitations. Characterization of the morphological PVAT adipocyte changes in our experimental paradigm with Western diet challenge and platelet depletion would be interesting to report. As PVAT is a primary contributor to vascular tone, determining the functional effect of the platelet’s manipulation of PVAT leukocyte content on blood pressure and blood vessel contractility are necessary future experiments to elucidate pathophysiological significance of this phenomenon [56,57]. However, the whole PVAT tissue was utilized for flow cytometry experimentation to gauge leukocyte composition of the entirety of PVAT, in line with the primary focus of this current study. By using the entire tissue, we avoided the potential misrepresentation of leukocyte proportions within PVAT if leukocytes were heterogeneously dispersed throughout the tissue. Additionally, we began to observe diet-dependent changes in PVAT leukocyte content with extended feeding and may have seen more significant dietary effects with a longer feeding duration. Regarding the administration of the platelet-depleting antibody negative control, usage of an isotype control antibody (instead of PBS, which was used in the current study) would have been ideal. Despite this, the routine utilization and characterization of the R300 antibody over several decades allow for the reasonable conclusion that our observations were due to the platelet-depleting effects of the antibody and not some unforeseen consequence, such as disruption of the bone marrow microenvironment [58,59,60,61]. Moreover, we only conducted platelet-depletion treatments proximal to PVAT collection, regardless of diet duration. Thus, the effect of early platelet depletion in an extended feeding regimen is unknown and it remains to be seen whether the effects of platelet depletion persist or whether endogenous restoration of the platelet population, in turn, restores PVAT composition to that of the vehicle-treated groups. Furthermore, our study was conducted using only male mice, and therefore, we cannot deduce whether our observations are universal or whether there are sex-specific difference in platelet-mediated PVAT manipulation. Undoubtedly, future investigations will require the utilization of both male and female mice to determine what impact sex may play in the platelet’s modulation of PVAT leukocyte content and function. Likewise, the impact of platelet depletion on PVAT functionality requires further investigation, as certain leukocytes, such as macrophages specifically, are known drivers of PVAT function [17,62,63]. Finally, the underlying mechanism of platelet-directed recruitment/denial of specific leukocytes to PVAT remains unknown. Platelets are capable of driving leukocyte recruitment by direct adhesion or indirectly via chemokine secretion, and the exact mechanism used to govern PVAT leukocyte residence necessitates further exploration that was out of the scope of the current study; thus, future studies using genetically modified mice with platelets that have impaired leukocyte binding/recruitment capabilities are also warranted.

Despite the limitations of this study, the results constitute the first ever report, to our knowledge, that platelets are key regulators of PVAT leukocyte content. Our findings indicate that platelets facilitate the residence of several leukocyte subsets within PVAT, and this influence has a temporal component that alters platelet-mediated recruitment as dietary stimuli transition from acute to prolonged, chronic insults. Moreover, platelets demonstrated a strong preference toward shifting the subsets of monocytes and macrophages. Additionally, the robustness of the platelet’s influence was tissue-specific, where tPVAT demonstrated a higher susceptibility to platelet regulation. Furthermore, our findings highlight the importance of platelets in relation to PVAT, and also that PVAT content is highly malleable, as platelet depletion rapidly induced changes in leukocyte composition. These results constitute an important first step in establishing platelets as a critical avenue of exploration for further understanding the mechanisms and consequences underlying PVAT cellular remodeling in health and disease.

## 4. Materials and Methods

### 4.1. Mouse Model and Diet Administration

Male C57BL/6J mice obtained from Jackson Laboratory (strain #000664) were implanted with microchip identifiers and randomized into groups fed ad libitum a control (Research Diets Inc., New Brunswick, NJ, USA, D12450J; 7% kcal sucrose, 10% kcal fat, 0.24% sodium) or Western (Research Diets Inc. D06111701; 30% kcal sucrose, 40% kcal fat, 8.0% sodium) diet. Two different cohorts of mice were utilized in the acute (2 weeks) and extended (8 weeks) feeding aims of this study. For the acute-feeding aim, mice were 10 weeks old and administered a diet for 2 weeks. Body mass was recorded weekly and body composition was assessed using EchoMRI (ECHO, Houston, TX, USA, Model# EMR-035) upon receipt (10 weeks old) and prior to euthanasia (12 weeks old). For the extended aim, mice were 4 weeks old and administered a diet for 8 weeks. Body mass was recorded weekly and body composition was assessed upon receipt (4 weeks old), after 4 weeks of feeding (8 weeks old) and prior to euthanasia (12 weeks old). Thus, for all PVAT analyses, both cohorts were studied at 12 weeks of age. The cohorts were additionally randomized into groups to receive either a vehicle or platelet-depleting antibody treatment. In sum, each cohort contained 4 groups: control diet–vehicle, control diet–antibody, Western diet–vehicle and Western diet–antibody, with 10 mice allocated per group for both the acute and extended aims. The Institutional Animal Care and Use Committee at the University of Arkansas for Medical Sciences approved all experimental procedures.

### 4.2. Platelet Depletion

Platelet depletion was achieved by intravenous administration of a commercially available monoclonal antibody (Emfret Analytics, Eibelstadt, Germany, R300) targeting platelet GPIbα to induce thrombocytopenia [64,65,66]. Antibody treatments were administered through the retro-orbital sinus at a dosage of 2 μg of antibody per 1 g of animal body weight with control treatments utilizing an equal volume of the vehicle (PBS) solution. Validation of platelet depletion was determined by quantification of platelet counts with an Abraxis Vetscan HM5C Hematology Analyzer (Allied Analytic 790-0000, Tampa, FL, USA) using whole blood collected from the retro-orbital sinus (Appendix A) in a separate, preliminary cohort of male mice. One injection maintained thrombocytopenia for 5 days, so 2 total treatments were utilized to maintain platelet depletion for 10 days, which coincided with the final 10 days of the feeding paradigm. Treatments were initiated so that PVAT collection would occur on the 10th day of platelet depletion.

### 4.3. PVAT Collection and Processing

Mice were euthanized via CO_2_ inhalation followed by a cardiac puncture and exsanguination. Both the abdominal and thoracic aortas with intact PVAT were removed and PVAT was dissected from its blood vessel under a microscope. These aortic PVAT depots were chosen, as they provided the largest tissues for sampling due to their proximity to the largest blood vessel (aorta). PVAT was digested using 0.5 mg/mL Liberase (Millipore Sigma 5401020001, St. Louis, MO, USA) and 0.01 mg/mL DNAse (Millipore Sigma, DNAse I 11284932001) in RPMI for 1 h at 37 °C. The cell suspension was filtered through a 40 μm cell strainer, centrifuged at 300 rcf for 5 min and resuspended in 1 mL complete RPMI. Cells were counted with an automated counter (Vi-Cell XR Cell Viability Analyzer, Beckman Coulter, Brea, CA, USA), with 4.0 × 10^6^ cells utilized for subsequent flow cytometry sample preparation. A diagram outlining the timing of all interventions, culminating with euthanasia and PVAT dissection, is presented in Appendix A.

### 4.4. Flow Cytometry Preparation

Cells were pelleted by centrifugation at 300 rcf for 5 min and supernatant discarded. Samples were stained by reconstituting in Zombie-Aqua LIVE/Dead dye diluted (1:360) in MACS Buffer (PBS w/0.4% EDTA and 0.5% BSA) for 10 min at room temperature. Afterward, samples were washed by adding 500 μL of MACS buffer, centrifuging and discarding the supernatant. Samples were then reconstituted in Block buffer (MACS + 1:100 FC Block [BD Biosciences 553141, Herlev Denmark] and 1:500 Rat IgG [Millipore Sigma I8015] and left to incubate for 10 min at 4 °C. Samples were washed and reconstituted in MACS buffer containing extracellular staining panel antibodies (Appendix A). Samples were incubated for 30 min at 4° C. Samples were washed, and then fixed and permeabilized using a Foxp3/Transcription Factor Staining Kit (Thermo Fisher 00-5523-00, Waltham, MA, USA) in accordance with the manufacturer’s protocol. Following fixation and permeabilization, samples were washed and reconstituted in a kit Wash buffer containing intracellular staining panel antibodies (Appendix A). Samples were incubated for 30 min at 4° C, then washed with a kit Wash buffer and again with PBS. Samples were reconstituted in 75 μL of PBS and passed through a 40 μm cell strainer 30 min prior to data acquisition to disrupt aggregates that potentially formed during flow cytometry preparation process.

### 4.5. Flow Cytometry Data Collection and Analysis

Prepared samples were processed using a Cytek Northern Lights full spectrum flow cytometer (Cytek Bio, Fremont, CA, USA). The entirety of the sample volume was consumed for data acquisition, and subsequent analysis was conducted using FlowJo^TM^ software (v10.9.0). The gating strategy used to identify broad leukocyte population is outlined in Appendix A. The gating strategies for myeloid and T cell populations are detailed in Appendix A. Data were plotted using GraphPad Prism^TM^ software (v10.2.3), which was also utilized for statistical analysis of all experimental data. All data are presented as the mean with standard error.

### 4.6. Statistical Analysis

Utilizing GraphPad Prism^TM^ (v 10.2.3), a 2-way ANOVA was used for the comparison of all datasets obtained for all groups, including both body/fat/lean mass measurements and PVAT leukocyte population metrics. Additionally, a Tukey’s post hoc comparison was performed to evaluate statistical differences between individual groups. A *p* (value) < 0.05 was used to determine statistical significance.

## 5. Conclusions

The novel results of our study demonstrate that (1) dietary influence required prolonged exposure before PVAT remodeling manifested and (2) platelets served as significant gatekeepers of PVAT leukocyte composition. Specifically, the removal of the platelet population caused rapid changes to the monocyte/macrophage population within PVAT. To our knowledge, this is the first study to demonstrate that platelets govern PVAT leukocyte composition, and, in turn, provides a novel avenue for further mechanistic investigation into how PVAT cellular content is remodeled during the onset and progression of diet-induced tissue dysregulation.

## Figures and Tables

**Figure 1 ijms-26-01625-f001:**
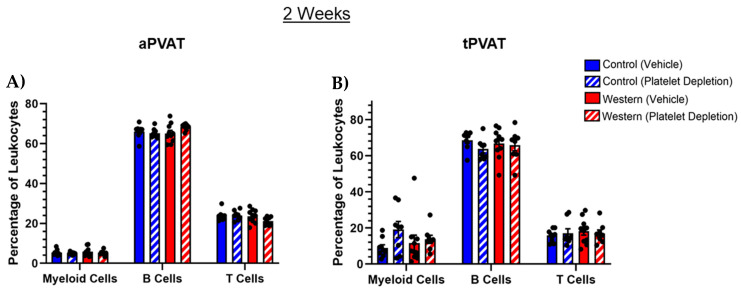
Analysis of broad leukocyte subsets in (**A**) aPVAT and (**B**) tPVAT following acute feeding and platelet depletion. Neither acute dietary modification nor treatment resulted in discernable changes when observing broad leukocyte classes in either PVAT depot. Leukocyte populations were identified by flow cytometry with pre-gating on the total, live, singlets and then CD45^+^ hematopoietic cells. Amongst CD45^+^ cells, myeloid cells were identified by CD11b positivity, T cells by CD3 positivity and B cells by B220 positivity. The percentage of cells are shown as a percentage of CD45^+^ cells.

**Figure 2 ijms-26-01625-f002:**
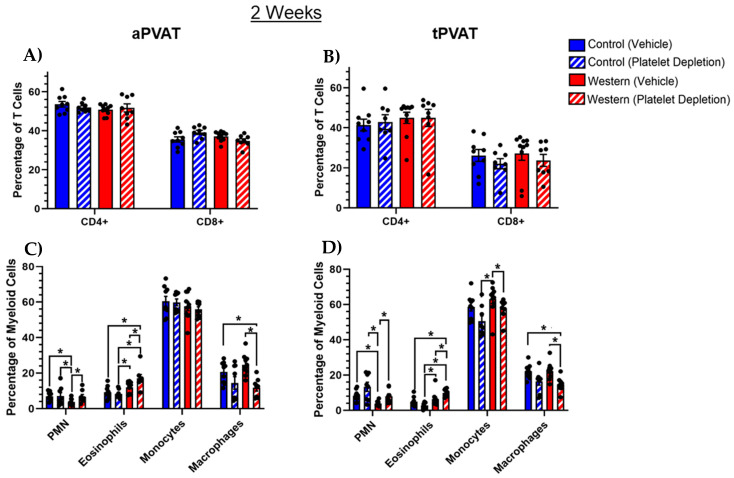
Investigation of specific leukocyte subsets after acute feeding (2 weeks) and platelet-depleting antibody treatments. The proportions of CD4^+^ and CD8^+^ T cells relative to the total CD3^+^ T cell population remained unchanged in (**A**) aPVAT and (**B**) tPVAT, regardless of Western diet or platelet depletion. Within the subsets of CD11b^+^ myeloid cells, platelet depletion facilitated a trend toward increased Ly6G^+^ neutrophils (PMN) and reduced CD64^+^MERTK^+^ macrophages in both (**C**) aPVAT and (**D**) tPVAT. Platelet depletion trended toward a reduction in Ly6C^+^ monocytes in tPVAT only, while the combination of a Western diet and platelet depletion increased the percentage of Siglec-F^+^ eosinophils in both depots. For aPVAT and tPVAT myeloid cells, the percentage of myeloid cell subsets (i.e., neutrophils, eosinophils, monocytes and macrophages) are shown as a percentage of CD11b^+^ cells. The percentage of CD4+ and CD8+ T cells are shown as a percentage of CD3^+^ cells. * *p* (value) < 0.05.

**Figure 3 ijms-26-01625-f003:**
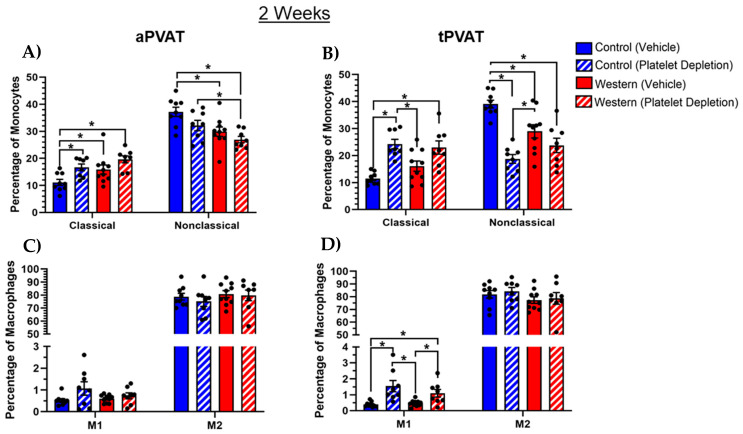
Alterations in monocytes and macrophages that resulted from acute diet and platelet-depleting treatments. (**A**) Within aPVAT, an intact platelet population paired with a control diet reduced Ly6C^hi^CX3CR1^lo^ classical and increased Ly6C^lo^CX3CR1^hi^ nonclassical monocytes, while platelet depletion paired with a Western diet reduced nonclassical monocytes. (**B**) Platelet depletion, regardless of diet, increased the percentage of classical monocytes and trended toward a reduction in the nonclassical monocytes in tPVAT. (**C**) Neither diet nor platelet depletion impacted the proportion of M1 or M2 macrophages within aPVAT. (**D**) Platelet depletion increased the proportion of M1 macrophages independent of diet. Regarding M2 macrophages in tPVAT, diet and treatment interventions had no effect. Monocytes and macrophages were identified by flow cytometry with pre-gating on the total, live, singlets and then CD45^+^CD11b^+^ cells. Amongst CD11b^+^Ly6G^−^Singlec-F^−^ cells, classical monocytes were defined as Ly6C^hi^CX3CR1^lo^ and nonclassical monocytes were defined as Ly6C^lo^CX3CR1^hi^. Amongst CD11b^+^ cells, the macrophages were defined as CD64^+^MERTK^+^ cells. * *p* (value) < 0.05.

**Figure 4 ijms-26-01625-f004:**
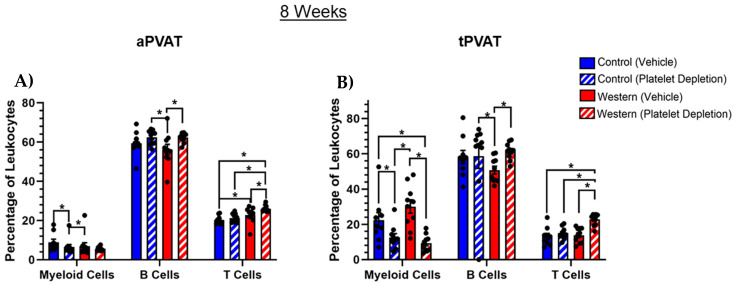
Leukocyte subset disruption in aPVAT and tPVAT following 8-week feeding and platelet depletion treatments. (**A**) Within aPVAT, the combination of a Western diet and platelet depletion significantly increased the percentage of CD3^+^ T cells. A control diet and intact platelets trended toward increasing the percentage of CD11b^+^ myeloid cells, while platelet depletion alone trended toward increasing the percentage of B220^+^ B cells. (**B**) Platelet depletion significantly decreased the CD11b^+^ myeloid cell population within tPVAT, independent of diet. Additionally, a Western diet paired with platelet depletion increased the percentage of CD3^+^ T cells, while a Western diet alone trended toward reducing B220^+^ B cells. * *p* (value) < 0.05.

**Figure 5 ijms-26-01625-f005:**
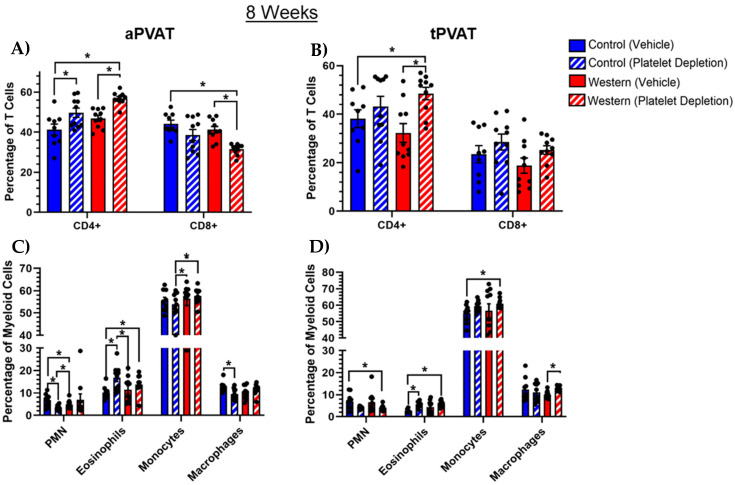
Lymphoid and myeloid cell subsets with 8-week feeding and platelet depletion treatment. (**A**) Within aPVAT, platelet depletion independent of diet resulted in an increase in CD4+ T cells, though the trend was more robust when paired with a Western diet. Inversely, the CD8+ population was diminished with platelet depletion coupled with a Western diet, with platelet depletion + control diet showing a similar trend. (**B**) Platelet depletion trended toward increasing the CD4+ population in tPVAT, withe Western diet pairing appearing more robust. Additionally, platelet depletion seemingly trended toward increasing the CD8+ population as well but did not yield a high degree of significance. (**C**) Platelet depletion coupled with a control diet increased the percentage of Siglec-F^+^ eosinophils in aPVAT. (**D**) Platelet depletion trends toward reducing the percentage of Ly6G^+^ neutrophils (PMN) in tPVAT. Additionally, an intact platelet population, with control diet feeding appeared to reduce tPVAT Siglec-F^+^ eosinophils. * *p* (value) < 0.05.

**Figure 6 ijms-26-01625-f006:**
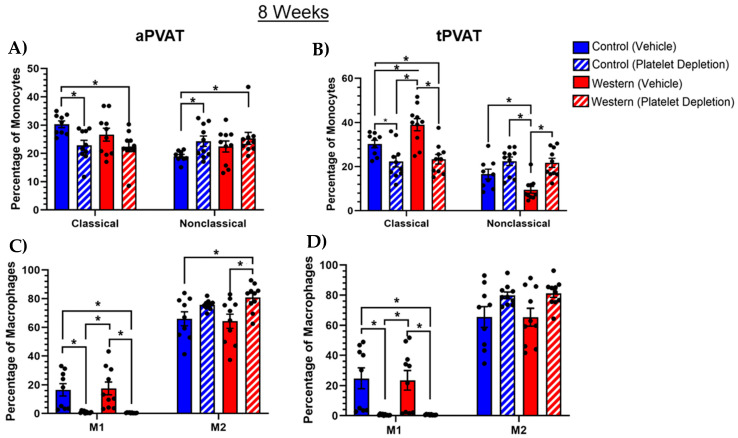
Frequency of monocyte and macrophage subsets in PVAT after 8-week extended feeding in conjunction with platelet depletion. (**A**) Control diet feeding coupled with intact platelets increased the percentage of Ly6C^hi^CX3CR1^lo^ classical and conversely decreased the percentage of Ly6C^lo^CX3CR1^hi^ nonclassical monocyte subsets in aPVAT. Platelet depletion weakly trended toward a reduction in the percentage of classical monocytes. (**B**) Platelet depletion alone decreased the percentage of classical monocytes in tPVAT. Western diet alone reduced the percentage of nonclassical monocytes, with platelet depletion eliciting a trend toward increased nonclassical monocytes. Platelet depletion, regardless of diet, significantly reduced the percentage of CD11b^+^CD64^+^MERTK^+^iNOS^+^ M1 macrophages in both (**C**) aPVAT and (**D**) tPVAT. Inversely, platelet depletion trended toward increasing the percentage of CD11b^+^CD64^+^MERTK^+^CD206^+^ M2 macrophages in both adipose depots. * *p* (value) < 0.05.

## Data Availability

The dataset outlined in this manuscript is available from the corresponding author upon request.

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
