# Peer review of "Platelets Modulate Leukocyte Population Composition Within Perivascular Adipose Tissue"

_ijms, 2025, doi:10.3390/ijms26041625_

Round 1
Reviewer 1 Report
Comments and Suggestions for Authors
This study aimed, apparently for the first time, to identify potential roles for peripheral blood platelets in leukocyte recruitment to perivascular adipose tissue (PVAT), as a function of conventional or Western diet in C57Bl/6 male mice. The authors used antibody-mediated platelet depletion coupled with altered diet and flow cytometry to analyze leukocyte compositions in PVAT. Data are presented indicating altered leukocyte recruitment to PVAT in these mice due to platelet antibody treatment, rather than Western diet, and that effects differed between short- and long-term treatment.
The manuscript is well written. The overall novelty of the study is providing a potential new link between blood platelets and inflammatory state in PVAT, which is important for vascular health. This is essentially a descriptive study that starts and ends with making this link. The study is essentially limited to flow cytometric analysis of PVAT; thus, mechanisms of how platelets may influence the observed outcomes are not explored, and there is no indication that the observed changes have functional effects. While the essential finding is of potential interest, the impact is therefore likely to be limited. Overall concerns are detailed below:
-
There is a major concern regarding the platelet depletion approach, which is paramount to the whole study. Antibody depletion of platelets is very well established to be transient, following a standardized approach which is employed here. Platelets will generally be reduced by up to 90% within 24 hours of treatment, but new platelet generation will restore the circulation platelet population by ~20% per day, resulting in fully restored platelet counts by 5 days. A delay in platelet recovery, if true, could indicate severe bone marrow destruction. The flattened curve of near-total platelet depletion from 1 hr through 96 hours (with no discernable variance, for n=5), followed by an immediate 50% restoration as shown in Supplementary Figure 1, is difficult to understand, as it appears to be quite unlike countless other similar studies. A cogent explanation is needed. Moreover, if the authors did indeed observe this highly unusual platelet recovery profile in this ‘preliminary cohort’, it is conceivable that platelet recovery followed different (and more conventional) dynamics in the PVAT study animals, but this was not tested. Additionally, the authors used PBS as a negative control, whereas equivalent doses of irrelevant antibodies would be the more appropriate control.
-
Physiological effects of altered leukocyte recruitment due to platelet depletion are not explored.
-
Studies to investigate a mechanistic basis for platelet modulation of leukocyte recruitment to PVAT are lacking.
-
Effects on leukocyte recruitment appear rather minor, by percentage. Considering this and the previous two points, which the authors acknowledge, these issues significantly limit the overall impact of the study.
-
Relatively minor point: only male mice were used. On the one hand this is commendable as it removes sex as a biological variable from the study. On the other hand, potential impact of sex differences is not explored. The authors are suggested to add this caveat to the discussion.
Reviewer 2 Report
Comments and Suggestions for Authors
The study aims to evaluate the role of platelets in leukocytes population remodeling in PVAT under normal or western diet. The manuscript is clear and relevant to field with well structure presented. The citing references are updated with reproducible results based on details given in method section. The figures and tables of both main manucript and supplementry are clear. The conclusion is coherent and consistent with the displayed evidences .
Minor comment:
- please add the citing reference for platelets depletion (2.2).
-Please add the post comparison test that has been used with 2 Ways ANOVA.
